# Analysis of the Aggregate Production Process with Different Geometric Properties in the Light Fraction Separator

**DOI:** 10.3390/ma15124046

**Published:** 2022-06-07

**Authors:** Tomasz Gawenda, Agnieszka Surowiak, Aldona Krawczykowska, Agata Stempkowska, Tomasz Niedoba

**Affiliations:** Faculty of Civil Engineering and Resources Management, AGH University of Science and Technology, Al. Mickiewicza 30, 30-059 Kraków, Poland; gawenda@agh.edu.pl (T.G.); aldona.krawczykowska@agh.edu.pl (A.K.); stemp@agh.edu.pl (A.S.); tniedoba@agh.edu.pl (T.N.)

**Keywords:** inward processing, aggregate, grain size, grain shape, light fraction separator (SEL)

## Abstract

This article presents an analysis of separation results in a specially designed and activated light fraction separator used to remove impurities from mineral aggregates. Laboratory tests conducted on a quarter-technical scale involved performing experiments to ascertain the scope for adjusting the variable settings of the separator operating parameters. These include the frequency and amplitude of pulsation, the height of the heavy-product reception threshold, the size of water flow and variations in the grain size and shape of the feed. During the experiments, the degrees of chalcedonite and dolomite grain purification were studied within the range of grain size for the feed: (2.0–4.0 mm for small grains, 8.0–16.0 for coarse grains and 2.0–16.0 mm for a wide range of grain sizes). The effects of the separator were assessed based on the amount of organic impurities in each heavy product. In all experiments, very good results were obtained, because the percentage of impurities in the product after separation was below 1% in accordance with the assumed technological standard assumption. Regarding the obtained content of light impurities with the separator set to optimal operating parameters, the percentage of light impurities in the product content was reduced to below 0.1%, which meets the guidelines described according to applicable standards. Multi-variant analysis allowed the optimal operating ranges of the separator to be determined, producing refined aggregate in terms of grain size and shape. The final results were also linked to the performance of the device, and its model dependencies were also determined.

## 1. Introduction

Aggregates are one of the most common materials used in engineering and infrastructure projects. Both fine and coarse aggregates accumulate in the region of more than 85% of cement and asphalt concrete [1]. The accumulative percentage of aggregates used in underlying layers is more than 90%. Due to its advantages, the demand for concrete is expected to grow, and it is estimated that this demand will double within the next 30 years [2].

The production and consumption of aggregates in Europe (including countries associated with the European Aggregates Association UEPG) currently amounts to over 3 billion tons per year. The vast majority of aggregates produced in the EU come from natural deposits (around 86.6%). The remainder of these aggregates are recycled (approximately 9.3%), artificial (approximately 2%) and extracted from marine areas (approximately 2%) [3]. In Poland, the production of natural aggregates in 2020 amounted to 256.8 million tons [4], which necessitates permanent access to aggregate deposits for the construction of road and rail infrastructure. Elements of embankments and other structural components (e.g., road foundation, gravel piles and abrasion layers) for which aggregate is used must have the appropriate quality and properties and must also meet demanding requirements. 

This always depends on individual cases; however, in general, sands and gravels with a low dust content are considered to be the best. In addition, the use of aggregates for the production of concretes and prefabricated elements as well as applications for general and hydrotechnical construction constitute a small but important share in the use of sands and gravels with appropriate physical and mechanical parameters for a given process. For this reason, the increasingly stringent requirements for the quality of mineral raw materials—in the case of aggregates, the highest possible degree of recovery (for gravel and sand), their reuse and the need to dispose of impurities—resulted in the search for new separation methods or improvements to existing ones [5,6,7,8,9]. 

Undesirable contaminants are lighter or heavier than the commercial product, and thus they can easily be separated using special devices called swords (or troughs) and drum scrubbers [10]. Aggregates can also be rinsed in a rod mill drum, drum screen or centrifugal pumps—or by using gravity methods [11]. Rinsing is a method of refining aggregates that enables dust impurities to be removed, i.e., grains of up to 0.05 mm in size. In aggregate production plants, they are used to remove light organic impurities found in aggregates. 

On the other hand, sand used for mortars or concrete must be rinsed naturally or in aggregate processing installations. Thus, bearing in mind the high demand for good quality aggregates, the development of rinsing methods in the pulsating stream of the water medium has been observed in recent years. To introduce this practice, well-known enrichment methods have been implemented in water-settling machines for cleansing gravel-sand feeds of the organic and mineral impurities that occur in them [12,13]. Experimental work has shown that the use of typical settling machines to enrich various materials, including aggregates, can be troublesome due to slight differences in the densities of the fractions separated [14,15]; in addition, it is best to separate narrow grain size fractions to eliminate the phenomenon of equally settling grains [16,17,18]. 

However, the results of research in this area confirm the suitability of settling machines for separating gravel-sand feed with a granulation of 2(0)–16 mm and for the removal of organic and mineral impurities from the aggregates obtained [19,20,21,22]. The use of enrichment technology in the settling machine enables organic (lignite and wood) and alkaline (silicates and horn) impurities to be removed from mineral aggregates. Contaminants of intermediate density can be separated from the sand and gravel in the settling machine due to the flat grain shape of the undesired substance, which causes the grains accumulate in the upper layer of the deposition bed [23]. 

For this reason, the feed of aggregates for separation in a pulsating medium stream should show variations in terms of geometric features especially size and shape [24,25,26], which enables the shape of the grains to be defined in many ways [27,28,29,30]. Purified grains of regular shape have an additional effect on the strength of prefabricated elements, e.g., concrete or thermal, such as dolomite [31]. Regular shapes can be achieved using the appropriate grinding operations and mechanical classification when preparing an aggregate for enrichment in special technological systems [32,33].

Furthermore, the question of particle shape for various types of aggregates is commonly an important topic of many scientific works. Spheroidal and irregular particles and their properties as they affect applications and processing have been discussed in [34,35,36]. The elongation and flatness of particles were discussed in [37]. The relations between the size and shape and shear characteristics of aggregates have been presented and examined in [38,39]. Modeling and predicting various features of the aggregates analyzed has been proposed in [40,41]. 

Construction sands and gravels are the most popular materials used for general construction and road construction. The directions for use of dolomite aggregate in construction, road engineering, gardening and others are also generally known, while chalcedonite is often used as road aggregate and filter grits. However, for several years, research has been conducted to demonstrate narrower, more specialized applications, in areas such as water treatment technology; as a silica filler for the production of paints, varnishes, putties, concretes; or in wastewater treatment technology. 

Waste chalcedonite fractions are also a valuable raw material for perlite plastic. Chalcedonite has great potential for use in sanitary engineering, as a sorbent for the removal of oil spills or filter filling during the last stage of wastewater treatment, substrates for horticulture and many others [42]. Considering all these aspects, these types of aggregates are also subject to treatment by various methods to increase their usable potential.

The aim of this work is to demonstrate the possibility of cleaning mineral aggregates by separation in a pulsating water medium, in a specially constructed light fraction separator (SEL) built on a quarter-technical scale. Variable parameters influencing the effects of the SEL separator include the properties of the feed (grain size and shape), the motor factors for operating the separator, including the flow rate of water to the separation as well as the amplitude and frequency of pulsation and design parameters (including the height of the collection threshold for the aggregate purified).

Generally, the main goal was to analyze the process of cleaning aggregates from a light fraction inside an innovative installation. As part of the research conducted, a laboratory-scale installation was created to produce mineral aggregates with regular and irregular grains in narrow fractions and clean these aggregates from light (organic) impurities in accordance with two patented inventions (Nos. PL233689B1 and PL233318B1). The results obtained enabled the development of design and process guidelines for the production of an industrial scale installation located in the mine. Research into this installation is currently being conducted.

## 2. Methodology

### 2.1. Materials and Methods

The aim of the experimental research was to investigate the possibility of purifying mineral aggregates using a specially constructed light fraction separator—SEL—built within the laboratory. The materials used for the tests were regular, irregular grains and a mixture of 50% of regular and irregular grains, designated as a feed made of a properly prepared aggregate with organic impurities added (particles of wood, bark, roots and charcoal). The percentage of impurities in the feed was 2%, and the aggregates used for research at work were chalcedonite and dolomite. The cleaned aggregates were in a various range of grain sizes, from a narrow fine fraction of 2.0–4.0 mm, through a coarser product in a wider grain size range of 8.0–16.0 mm to a full grain size range of 2.0–16.0 mm.

The preparation of regular and irregular aggregate for testing was conducted in a special technological system in accordance with Patent No. PL233689 and PL233318B1. From the material that was crushed in the jaw crusher, narrow grain fractions were separated in a vibrating screen and then directed to another screen with a slotted sieve (with appropriately matched sieve slots). As a result of classification, the required grain fractions were obtained in the over-sieve product (with regular grains) and in the under-sieve product (with formless grains). The assessment of the grain shape was conducted in accordance with the applicable standard for the production of aggregates PN-EN 933-3:2012 [43], which defines regular (conventional) grains as those whose length does not exceed three times their width and thickness. 

Similarly, materials were prepared for separation in the full grain size range, i.e., 2.0–16.0 mm. The exact scheme of preparation for selected fractions of regular and irregular grains is shown in publications [16,44]. Achieving the main research objective made it possible to define guidelines for the regulation of certain work parameters for the newly constructed prototype of a light fraction separator—first in a laboratory scale and then in an industrial scale—to be used in the technological system. The objective of the experimental study was to analyze how each of these fractions would purify with the effect of grain shape and grain class range taken into consideration.

### 2.2. Experiment

The device in which the refinement of aggregates was conducted was a prototype developed by HTS Gliwice on a quarter-technical scale—a separator for a light fraction (SEL), whose appearance is shown in Figure 1. In the separator, the distribution medium is water, which is used as a working medium for cleansing aggregates of impurities, most of which are organic. The contaminated material is directed into a machine filled with water, in which a wave movement of water is created. 

As a result of undulation, contaminants are washed out and float on the surface of the water. They are then discharged through overflow threshold No. 1 (Figure 2) along with the water side outlet from the machine. The purified aggregate escapes with water from the front of the machine through overflow threshold No. 2. Sand contaminants fall out and are drained continuously through tubes from the bottom of the machine. The degree of aggregate purification varies depending on the amount of water used and the degree of aggregate contamination.

The operating parameters for the SEL separator are summarized in Table 1. The maximum capacity of the separator is 2750 kg/h, and it is driven by a 4 kW electric motor. The frequency of the pulse of the stroke is adjusted by changing the revolutions of the electric motor using a frequency inverter within a range of approximately 60 to 90 cycles/min. The separator has four polyurethane sieves with 2 mm openings, whose total length is 1160 mm, while their width is 150 mm, creating a bed working surface of 0.174 m^2^. 

The maximum capacity of the device reaches 2750 kg/h, and the maximum water requirement is up to 5.5 m^3^/h. The product collection system is equipped with two water tanks, which previously separate, from the water, the products of the heavy fraction (aggregates) on the sieve and the light fraction (organic impurities) on the bag filter. The water is collected in tanks and then pumped into the main feed tanks. In this way, the flow of water through the machine is a closed system, which significantly reduces water consumption and stops water from escaping from the system.

The aggregate refinement experiments performed in the SEL separator were conducted with a constant light fraction reception setting (threshold No. 1 equal to 15 cm) and various settings for the heavy fraction acceptance threshold (No. 2): a low threshold of 13.5 cm and two high thresholds of 16.5 and 18 cm. The water flow rates were set at 1.3; 1.8 and 2.1 dm^3^/s, and the pulsation frequencies were 60, 70, 80 and 90 cycles/min. The amplitude of pulsation was selected accordingly to the size of the separated grains (5 and 8 cm). After determining the conditions of water flow and the feeding rate as well as stabilizing separator operation, representative aggregate samples were taken within 10 s. 

The aggregate was then washed to determine the percentage of remaining impurities in the aggregate. The results presented in this work are preceded by preliminary tests, the results of which enabled the appropriate setting for the machine to be selected to make it possible to collect representative samples of the purified aggregate. Therefore, the grain fractions of chalcedonite and dolomite were selected and tested at the full range of frequencies and amplitudes of pulsations, flow intensities and heavy-product reception threshold settings. 

In some separator operating conditions, product collection was impossible due to incorrect machine settings for certain raw material grains and the transition of aggregate grains to a light product due to incorrect settings of pulsation frequency and the aggregate reception threshold. For dolomite aggregate, it was difficult to obtain irregular grains of a certain size due to the nature of the aggregate and the way it is crushed. Table 2 summarizes the characteristics of the tests that were considered relevant to the multivariate analysis of the impact of individual parameters on the degree of purification of aggregate from impurities.

The results of the research presented in the article, as well as the attainment of the main research goal, enabled guidelines to be defined for regulating certain operating parameters for a newly constructed prototype embedded machine on an industrial scale, to be used in a technological system according to Patent No. PL233318B1.

According to Patent No. (PL233689B1), the system is designed so that even with only one crusher (e.g., a jaw crusher), a final aggregate with no more than 2–3% irregular particles can be obtained. The system only requires the use of vibrating screens with square mesh and slotted mesh screens working together by pelting each other. The crusher is used during the first or second stage of operation. The purpose of the multi-deck screen is to divide the aggregate into narrow particle fractions, which go to a single-deck multi-product screen with a slotted screen, and then the irregular particles are screened out (lower product) and returned to the crusher. 

The irregular grains can be comminuted in the same crusher or at a secondary crushing stage, in a VSI crusher, for example. This will have a positive effect on the product quality. The content of irregular particles in the final product will depend on the efficiency of the slotted sieve and, in particular, on the proportion between the narrow range of particle fractions and the size of the slot in the sieve. The slotted sieve should be selected on the basis of 50–70% of the maximum grain size of the fraction. The idea of the aggregate production circuit, with closed recirculation for selective screening and crushing operations, was presented in more detail in another article [32].

## 3. Results and Discussion

### 3.1. Multivariate Analysis of the Influence of Variable Parameters on Aggregate Refining Effects

During the aggregate refining process, involving the removing organic impurities in a separator with a pulsating stream for the working bed in an aqueous environment, there are a number of variable factors that determine its course. They are related to both the properties of the feed, the machine’s operating settings and the hydrodynamics of the process, and they affect the final effects of this process, i.e., the production of a new, better quality product. Therefore, in order to illustrate how and to what extent the parameters analyzed determine the amount of separation in the demonstration model of the SEL light fraction separator, a multi-parameter analysis was conducted. 

Variables, such as the amount of impurities in the aggregate after separation (axis *y*), the pulsation frequency of the machine piston (axis *x*) and the flow rate of water to the separator (axis *z*) were taken into account for the analysis. A three-parameter analysis was performed, which, within a certain range of variability of individual parameters, makes it possible to forecast the results of the separation for individual variables. All possible circumstances affecting the results of the experiment were analyzed. The light pollutant content obtained with optimal SEL operating parameters was lower than 0.1%, which meets the guidelines of PN-EN 1744-1 [45]. 

Moreover, an additional milestone for the project, entitled “Elaboration and construction of a set of prototype technological devices to form an innovative technological system for aggregate beneficiation along with tests conducted in conditions similar to real ones”, was to reduce contamination in aggregates to 1%. Not all of the figures were taken into consideration and shown in 3D images in this paper. Nevertheless, their analysis enabled the following relations to be revealed. An analysis of all possible circumstances affecting the results of the experiment was performed. These are included in Table A1, Table A2 and Table A3 (Appendix A).

The analysis of the results made it possible to formulate general observations. For fine grains with a size of 2.0–4.0 mm, the best level of aggregate purification is achieved for low water flow rates below 2 dm^3^/s, which was obtained during the experiments. As far as irregular grains are concerned (Figure 3a), it is clear that the proportion of impurities in the product would be at their highest in the case of higher pulsation frequencies of 80–90/min and higher water flows of approximately 4–4.5 dm^3^/s.

In the case of regular grains (Figure 3b), the proportion of impurities in the product is low, regardless of the size of the water flow and the frequency of pulsation. The lowest pollutant values are observed for low water flows and pulsation frequencies of 70–75 and 90–95 cycles/min. However, for the smallest grains of the feed, the amount of impurities in the aggregate was also low, regardless of the water flow and pulsation. The smallest amount was obtained at low water flow rates and pulsation frequencies of 85–95 cycles/min. 

The shape of fine grains affects the degree of purification of the aggregate, and this is most noticeable in the case of grains not normally used in the experiments performed. Earlier studies confirmed these results, which showed that screening of feed material into narrow particle size fractions as well as the separation of these fractions in terms of shape brings more favorable results using pulsating movements to enrich aggregates. This is due to the narrowing of the parameters for the relevant aggregates and the elimination of equally settling particles [16,46].

In the case of irregular grains (Figure 4a), the amount of impurities in the product is at its lowest with a water flow of 2–3 dm^3^/s and pulsation frequencies of 75 and 85–95 cycles per minute. Variable operating conditions in the separator for regular large grains of 8.0–16.0 mm (Figure 4b) enable the lowest values of impurities to be attained with low water flow and pulsation frequencies of 85–90 cycles per minute, with little impact on the amount of impurities. When it comes to the conditions for separating the feed from coarse fraction grains, the amount of impurities in the product is by far the lowest for low flows and pulsation frequencies of 60–65 cycles/min. 

Very good results were obtained because the pure final product, i.e., the purified aggregate, basically did not contain impurities. At the same time, the best purification effects were achieved with a water flow of 1–2 dm^3^/s and pulsation frequency of 90–95 1/min. As the water flow rate increases, the content of impurities in the product increases. Lowering the aggregate collection threshold to 13.5 cm results in the lowest aggregate contamination in the case of water flow rates of 1.5–2 dm^3^/s and pulsation of 85 cycles/min. 

The appropriate selection of the characteristics of the water pulsation cycle for a given type of enriched coal or gravel-sand feed significantly affects the technological parameters of the process obtained [21]. Kowol and Matusiak [13] proved the degree of reduction of the proportion of impurities in the gravel product and the level of aggregate loss in the lightweight product is dependent both on the density of impurities as well as on the grain size distribution of the feed. According to changeable hydrodynamic parameters of separator, such as the amount of water in the separator, to ensure sufficient liberation of particles according to their geometrical properties and density [46,47,48,49].

For dolomite grains, the proportion of impurities in the aggregate after separation was at its lowest with a water flow of 1.5–2 dm^3^/s and a pulsation frequency of about 95 cycles/min (Figure 5a). If the aggregate collection threshold is raised, the amount of impurities in the product is at its lowest with a small water flow and a small number of pulsations of 55 cycles/min (Figure 5b).

### 3.2. Influence of SEL Separator Operating Conditions on Process Efficiency

The real-time performance of the machine is the result of many factors affecting the efficiency of material cleaning. These are important in achieving a high level of efficiency of aggregate cleaning regulation of process factors, such as pulsating motion. Two important parameters that significantly affect the pulsation characteristics are the separation efficiency and device performance [50].

Figure 6a–d shows the effect of pulsation frequency on separator efficiency for regular, irregular and overfeed grains at different settings of the values of structural and motion parameters. For regular grains of fraction 2.0–4.0 mm, the performance characteristics change slightly within the tested range of variation in pulsation frequency and flow rate (Figure 6a). It should be mentioned that the fine fraction experiments of 2.0–4.0 mm were performed at the receiving threshold No. 2 equal to 18 cm. For regular grains of the 8.0–16.0 mm fraction, the level of efficiency increases in line with pulsation frequency and also at a lower aggregate reception threshold (13.5 cm). 

SEL efficiency decreases at higher pulsation frequencies (80 and 90/min) and under the conditions of a maximum water flow rate of 2.1 dm^3^/s and an increased threshold No. 2 of 16.5 cm—Figure 6b. For irregular grains of the discussed grain fractions, the maximum level of efficiency was achieved for 70 and 90 pulsation cycles per minute with water flows of 1.3 and 1.8 dm^3^/s. At a maximum water flow of 2.1 dm^3^/s, the level of efficiency increased in line with the number of pulsations (Figure 6c,d) and with a higher threshold setting for a thicker fraction (16.5 cm). For fine-grade feed grains, the yield was similar to that of regular grains.

To sum up, it can be concluded that, depending on the characteristics of the grains fed into the separator (size and shape), the operating conditions for the separator should be properly selected to attain the best possible purification of natural and recycled aggregate grains from impurities [51]. It is appropriate to selectively refine grains of different shapes: regular and irregular, due to the fact that they require slightly different machine settings. In addition, the characteristics of the operating and working conditions for the SEL separator using a pulsating stream, which were presented in this paper, enable it to be used for recycled materials, construction and demolition waste. This also makes it possible for the separator to be used within the circular economy [52].

## 4. Conclusions

Experiments conducted in laboratory conditions, involving testing an SEL separator operating on a quarter-technical scale, were used for a detailed multifactorial analysis of the aggregate purification process in terms of variable factors of a structural, operational nature, in relation to the characteristics of the feed grains used. Subsequently, the results of the experiments formed the basis for developing guidelines and final conclusions regarding conditions for the aggregate refining process, depending on the characteristics of the feed and the settings of the separator:-The amplitude and frequency of pulsation should be appropriately selected according to the grain size of the material fed into the separator. As tests showed, coarse grains > 8 mm purify better at a higher amplitude of 8 cm and a pulsation frequency of 60–80 cycles/min, while fine grains < 8 mm reach a better level of purification at an amplitude of 5 cm and a pulsation frequency of 60–80 cycles/min.-It is important to set the height of the threshold of acceptance for the heavy fraction (No. 2), i.e., purified aggregate. In the case of refining fine grains < 8 mm, better results are obtained by working at a higher aggregate reception threshold (16–18 cm),-The total water consumption in the case of separation of impurities can be reduced to 1.3 dm^3^/s. Due to the fact that the size of the water jet does not affect the quality of the purified aggregate in any clear way, it is preferable to work at lower flow rates but with a higher setting of the aggregate reception threshold while maintaining higher pulsation frequencies of 80–90 cycles/min for finer grains. For coarse grains, excessive water flows and a high reception threshold cause turbulence in the reception range of the light product, and under such conditions, the aggregate grains are drawn into the light product, especially at low levels of pulsation frequency.

In wide grain size classes, e.g., 2.0–16.0 mm (and in regular and irregular grains mixtures), despite the high purity of aggregates achieved in the product, we noted that finer aggregates leaked into the waste light fraction causing losses. When the fractions were narrowed and separated by shape, no such losses were observed.

The results of the research presented in this article with the use of a laboratory light fraction separator (SEL) were used to formulate guidelines for the design and construction of a device designed to enable the production appropriate quality aggregates under industrial conditions. Such an innovative technological system was developed according to the patented invention (Patent No. PL233689) for the refinement of aggregates by grinding, screening and gravitational enrichment (separation). It was implemented in a technological system for producing and refining aggregates in a dolomite mine.

## 5. Patents

Two patents granted in Poland were utilized in the paper:Author: Gawenda, T. Title: Układ urządzeń do produkcji kruszyw foremnych, AGH w Krakowie, Patent No. PL233689 granted on 8 July 2019, http://patenty.bg.agh.edu.pl/pelneteksty/PL233689B1.pdf (accessed on 24 April 2022).Authors: Gawenda, T.; Saramak, D.; Naziemiec, Z. Title: Układ urządzeń do produkcji kruszyw oraz sposób produkcji kruszyw. AGH w Krakowie, Patent No. PL233318B1 granted on 7 June 2019, http://patenty.bg.agh.edu.pl/pelneteksty/PL233318B1.pdf (accessed on 24 April 2022).

## Figures and Tables

**Figure 1 materials-15-04046-f001:**
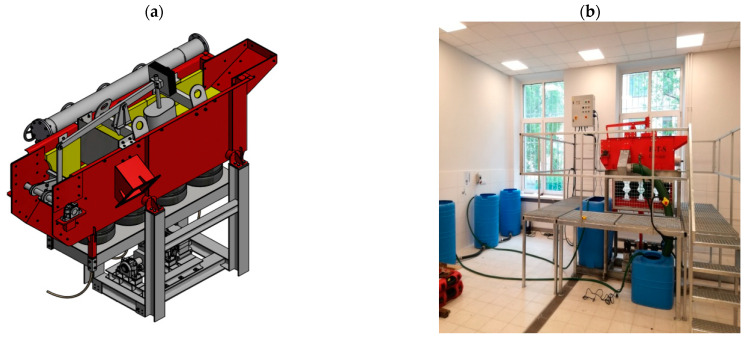
SEL light fraction separator developed by HTS Gliwice: (**a**) model and (**b**) device on a quarter-technical scale.

**Figure 2 materials-15-04046-f002:**
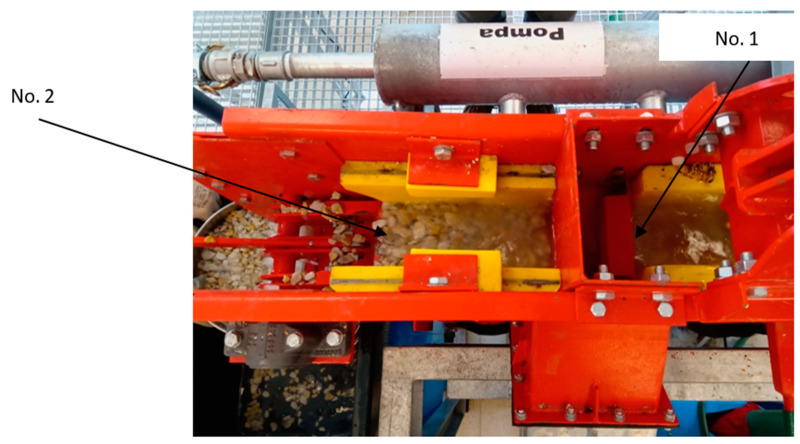
Arrangement of the thresholds for the reception of light fraction No. 1 and heavy fraction No. 2.

**Figure 3 materials-15-04046-f003:**
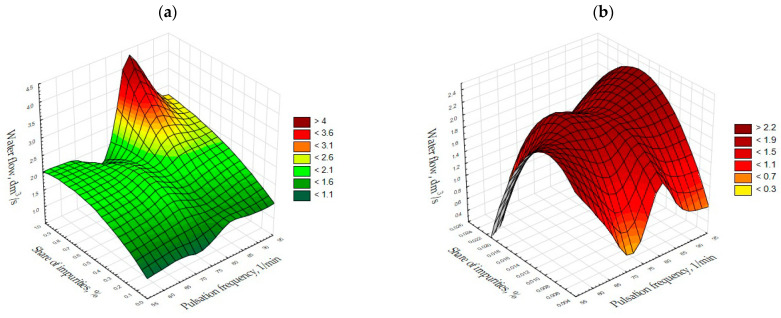
Chalcedonite, fraction 2.0–4.0 mm, amplitude 5 cm and threshold height 18.0 cm. (**a**) Irregular grains; (**b**) Regular grains.

**Figure 4 materials-15-04046-f004:**
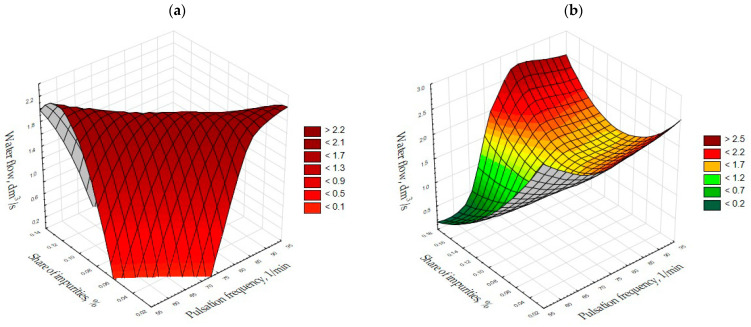
Chalcedonite, fraction 8.0–16.0 mm and amplitude 8 cm. (**a**) Irregular grains, threshold height 16.5 cm; (**b**) Regular grains, threshold height 16.5 cm.

**Figure 5 materials-15-04046-f005:**
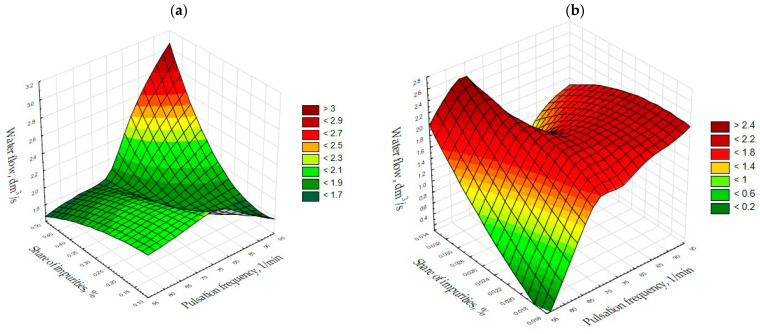
Dolomite, fraction 2.0–16.0 mm, regular grains and amplitude 8 cm. (**a**) Threshold height 13.5 cm; (**b**) Threshold height 16.5 cm.

**Figure 6 materials-15-04046-f006:**
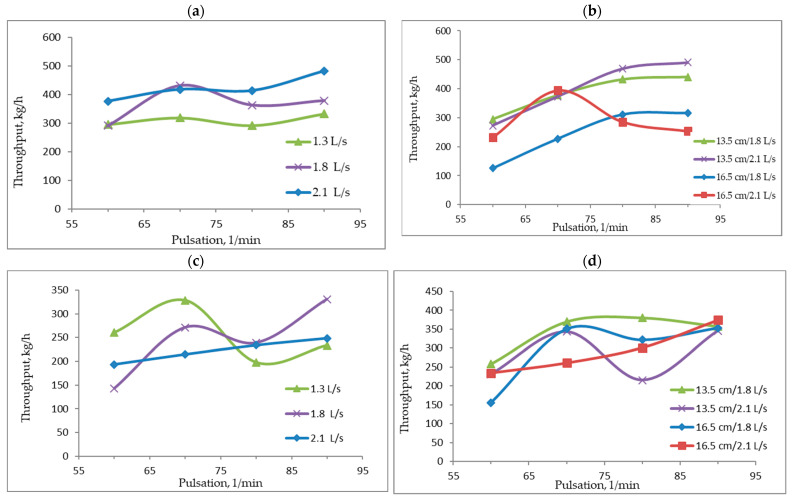
Influence of the pulsation frequency on the SEL performance for different design and operating conditions. (**a**) size fraction 2.0–4.0 mm, regular particles; (**b**) size fraction 8.0–16.0 mm, regular particles; (**c**) size fraction 2.0–4.0 mm, irregular particles; (**d**) size fraction 8.0–16.0 mm, irregular particles.

**Table 1 materials-15-04046-t001:** Characteristics of the SEL separator.

Name	Unit	Value
Maximum throughput	[kg/h]	2750
Maximum water flow	[dm^3^/h]	5500
Frequency of bellow pulsation	[1/min]	60–90
Bellow jump	[mm]	50–1400
Nominal power	[kW]	4.0
Sieve dimensions	[mm]	150 × 1160

**Table 2 materials-15-04046-t002:** Summary of the tests performed.

Test Number	Material Type	Particle Type	Particle Size [mm]	Threshold Setting No. 2 [cm]	Amount of Water [dm^3^/s]	Amplitude/ Stroke [cm]
1	chalcedonite	feed	2.0–4.0	18.0	1.3	5
					1.8	
					2.1	
2	chalcedonite	regular	2.0–4.0	18.0	1.3	5
					1.8	
					2.1	
3	chalcedonite	irregular	2.0–4.0	18.0	1.3	5
					1.8	
					2.1	
4	chalcedonite	feed	8.0–16.0	16.5	1.8	8
					2.1	
5	chalcedonite	feed	8.0–16.0	13.5	1.8	8
					2.1	
6	chalcedonite	regular	8.0–16.0	16.5	1.8	8
					2.1	
7	chalcedonite	irregular	8.0–16.0	13.5	1.8	8
					2.1	
8	dolomite	regular	2.0–16.0	13.5	1.8	8
					2.1	
9	dolomite	regular	2.0–16.0	16.5	1.8	8
					2.1	

## Data Availability

The data presented in this study are available on request from the corresponding author.

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
