# Peer review of "Analysis of the Aggregate Production Process with Different Geometric Properties in the Light Fraction Separator"

_materials, 2022, doi:10.3390/ma15124046_

Round 1

Reviewer 1 Report

Overall, a good article. Minor revisions are recommended – see my detailed comments below. Also in general, a bit more description of the methods and results would seem to be helpful to the readers of this general materials journal. The introduction is nice and helps with that task.

Abstract: Should not use acronyms (SEL) in the abstract. Why is 0.8 % a good result? The authors should say something like “0.8%, well less than the limit of 4% demanded by the appropriate standards” or something like that. I don’t know what the limit should be, I just used 4% as an example.

Introduction: “Production and consumption of aggregates in Europe” is this in concrete or overall use? 5% recycled aggregates is a small number, if referred to overall use, but may be reasonable for concrete. References?

I am not sure if the readers of this journal would know the definition of “equidiveness” – I didn’t. Should be briefly defined in the paper.

“For this reason, the transmission of aggregates for separation in the pulsating stream of the medium should be characterized by the diversity of  grains in terms of geometric features” this sentence is not clear – what is actually meant by “transmission” and “characterized by the diversity”?

Section 2.2: “grains to a light product due to, for example, poor settings of pulsation frequency and aggregate reception threshold” avoid words such as “poor” The settings may have resulted in materials that did not meet a certain criterion – say something specific like this, not general words such as “poor.” There may be other instances in the paper – they should be changed, too.

Why do the authors sometimes say “pollutants” and sometimes “impurities?” They are two very different things in English.

I suspect that Figs. 3-5, the 3D figures, are not sufficiently described for the reader to get meaningful insight from them. I suggest more description.

Section 3.2: “For irregular grains of the discussed grain classes, the maximum efficiency” How do the authors define “efficiency?” All I see on Fig. 6, for example, are the y-axis labels for throughput, kg/h. If efficiency is defined as the maximum throughput, then the authors should say so. But I am still not sure what efficiency means in this context. Efficiency = throughput /energy input?

The earlier reference to the patent may not be enough for the readers of this journal, since the patent appears to be in the Polish language. A little more description of the patented features would be good for the article.

Author Response

Thank You very much for the insightful and constructive summary of our manuscript and for the effort put into it. We present the points regarding the comments given by the Reviewer point by point and marked red colour into the text.

Reviewer 1

Overall, a good article. Minor revisions are recommended – see my detailed comments below. Also in general, a bit more description of the methods and results would seem to be helpful to the readers of this general materials journal. The introduction is nice and helps with that task.

Point 1. Abstract: Should not use acronyms (SEL) in the abstract. Why is 0.8 % a good result? The authors should say something like “0.8%, well less than the limit of 4% demanded by the appropriate standards” or something like that. I don’t know what the limit should be, I just used 4% as an example.

Response 1: The acronyms SEL has been removed from the summary and changed: “a maximum of 0.8%.” to “below 1% accordance with the adopted technological assumption. The obtained content of light impurities with optimal operating parameters of the separator was lower than 0.1%, which meets the guidelines contained in the standard”And “The obtained content of light pollutants with optimal SEL operating parameters was lower than 0.1%, which meets the guidelines of PN-EN 1744-1, and an additional milestone in the project was to reduce contamination in aggregates to 1%.” – we put to the section 3.1.

Point 2: Introduction: “Production and consumption of aggregates in Europe” is this in concrete or overall use? 5% recycled aggregates is a small number, if referred to overall use, but may be reasonable for concrete. References?

Response 2: We we updated the data and add references: Annual Report UEPG 2020-2021, (https://uepg.eu/mediatheque/media/Final_-_UEPG-AR2020_2021-V05_spreads72dpiLowQReduced.pdf).

Point 3: I am not sure if the readers of this journal would know the definition of “equidiveness” – I didn’t. Should be briefly defined in the paper.

Response 3: right the word was misused we put word “equal settling” instead “equidiveness”. That means: large and lighter grains, they fall at the same settling velocity as small grains, but with high density, it is recommended to direct narrow grain classes to the process in order to avoid equal settling grains.

Point 4: “For this reason, the transmission of aggregates for separation in the pulsating stream of the medium should be characterized by the diversity of  grains in terms of geometric features” this sentence is not clear – what is actually meant by “transmission” and “characterized by the diversity”?

Response 4: badly translated it has been changed to „For this reason, the feed of aggregates for separation in a pulsating medium stream should be characterized by grain differentiation in terms of geometric features, i.e. size and shape”

Point 5: Section 2.2: “grains to a light product due to, for example, poor settings of pulsation frequency and aggregate reception threshold” avoid words such as “poor” The settings may have resulted in materials that did not meet a certain criterion – say something specific like this, not general words such as “poor.” There may be other instances in the paper – they should be changed, too.

Response 5: “poor” has been changed to “incorrect”

Point 6: Why do the authors sometimes say “pollutants” and sometimes “impurities?” They are two very different things in English.

Response 6: It should be “impurities” and we have improved the text

Point 7: I suspect that Figs. 3-5, the 3D figures, are not sufficiently described for the reader to get meaningful insight from them. I suggest more description.

Response 7: We add: “Variables such as the amount of impurities in the aggregate after separation (axis y), the pulsation frequency of the machine piston (axis x) and the flow rate of water to the separator (axis z) were taken into account for the analysis.” In addition, the descriptions on the axes of the drawings have been enlarged to make them more legible.

Point 8: Section 3.2: “For irregular grains of the discussed grain classes, the maximum efficiency” How do the authors define “efficiency?” All I see on Fig. 6, for example, are the y-axis labels for throughput, kg/h. If efficiency is defined as the maximum throughput, then the authors should say so. But I am still not sure what efficiency means in this context. Efficiency = throughput /energy input?

Response 8: For the purpose of this work, the separator throughput was defined as the amount of obtained products after separation (aggregate and impurities) per time unit. The samples were collected after the conditions stabilized by the separator and converted to [kg/h].

Point 9: The earlier reference to the patent may not be enough for the readers of this journal, since the patent appears to be in the Polish language. A little more description of the patented features would be good for the article.

Response 9: The system according to patent nr. (PL 233689 B1) is designed so that even with only one crusher (e.g., a jaw crusher), a final aggregate with no more than 2–3% irregular particles can be obtained. The system requires only the use of vibrating screens with square mesh and slotted mesh screens cooperating in a pelting way in return with the crusher located in the first or second stage. The purpose of the multi-deck screen is to classify the aggregate into narrow particle fractions, which go to a single-deck multi-product screen with a slotted screen, and then the irregular particles are screened out (lower product) and returned to the crusher again. The irregular grains can be comminuted in the same crusher or in a secondary crushing stage by impact, for example, in a VSI crusher, which will have a positive effect on the product quality. The content of irregular particles in the final product will depend on the efficiency of the slotted sieve and, in particular, on the proportion between the narrow range of particle fractions and the size of the slot in the sieve. The slotted sieve should be selected based on 50–70% of the maximum grain size of the class. Idea of the aggregate production circuit with a closed recirculation for selective screening and crushing operations has been presented in more detail in the following article [Gawenda T., 2021, Production Methods for Regular Aggregates and Innovative Developments in Poland, Minerals 2021, 11, 1429. https://doi.org/10.3390/min1112142].

Reviewer 2 Report

The paper "Analysis of the aggregate production process with different geometric properties in the light fraction separator" presents the results of a study with a particle separator to be used by the aggregates industry. The conclusions obtained demonstrate that it is possible to select the best adjustment of the equipment for the characteristics of the material to be separated. In addition, two patents from one of the authors of the manuscript were used as a basis for the methodology.

The manuscript is clear and relevant.  The cited references are mostly recent publications. The abstract is well written and reflects well what was developed in the study. Therefore, I consider that no further adjustments are necessary.

Author Response

Thank You very much for your favorable review of our manuscript.

Reviewer 3 Report

Dear authors,

The article is well presented. However I believe that is very important improve the discussion of the results.

The discussion of the results is focused only on the results obtained. There is no deep discussion with previous results in the literature which are well presented in the introduction. We wait for the moment when the authors will justify the reason for the research with the results obtained and with the current literature. I understand that this item is fundamental to the article.

In general, I understand that the article presents the results of trials without a deeper discussion with a comparison with results from the literature to justify some character of scientific contribution and/or innovation. I understand that a more scientific analysis is needed to justify and illustrate the contribution of the research.

Author Response

(The authors gave the same response as above.)

Reviewer 4 Report

Analysis of the aggregate production process with different geometric properties in the light fraction separator

While the work presented in the paper make sense, however, there are many issues which need to be addressed first before the paper can be accepted for publication. I have highlighted these issues below.

  1. Not sure what is SEL. Please when you use any abbreviation, make sure you explain it in it first use so that the readers are not confused. Later at the introduction section, you have provided some meaning of the SEL, but it still does not make sense to me.

  1. The statistics presented in the first line of the introduction section needs references/ citations. I think it would be also appropriate to focus on the world rather than just Europe/ Poland. The best option would be to start from world, then narrow it towards Europe and Poland. For instance, Umar et al., (2019) in their paper “A Modified Method for Los Angeles Abrasion Test” argued that “aggregates are one of the most common materials used in engineering and infrastructure projects. Roughly, both fine and coarse aggregates accumulate more than 85% of cement and asphalt concrete. The accumulative percentage of aggregates used in underlying layers is more than 90%”. Likewise, Umar et al., (2020) in their paper “Developing a Sustainable Concrete Using Ceramic Waste Powder” discussed that “Due to several advantages, a concrete requirement is expected to be growing and it is estimated that this requirement will become double in the next 30 years”. I think we you will include such arguments with the above references will make your paper more reasonable and convincing as introduction is very important part of a research paper, and if the argument here are not compelling, the readers will not take interest in the paper.

  1. The introduction also includes some relevant literature, and as there is no specific heading on literature review, I suggest to change the introduction heading to “Introduction and Literature review”.

  1. While you have provided the aims of the research at the end of the introduction section, but I still do not see strong compelling arguments for this research. apart from the providing the purpose of the research, you need to tell the readers (in a more convincing manner) the reasons/ implications of the proposed research – may be considering practical/industrial and or educational/research implications.

  1. Any specific reasons for using only chalcedonite and dolomite – does this reduce the generalization of findings. Were there any scientific reasons for this. Are these the only aggregates which are commonly used in Poland. It is important to answer these questions with clarity so that the readers can consider your research implications and also understand the reasons for selecting only these types of aggregates.

  1. Overall, the research approach is poorly described with almost no justifications and reference to relevant literature. Some sentences are vague and do not make sense. For instance, the second line of methodology section. There is no clarity on organic impurities. Were these already included in sample or were added to the sample. More information on the PL233689 as I do not see an accessible link in the reference provided. This is also the case with PL 233318B1.

  1. The tables/ figures should be reported in the paper in the sequence as they are quoted in the text.

  1. The results and discussion should be reported with clarity. The flow needs to be improved here. Too many figures are there which make it difficult to read/ understand. This also make the paper unnecessary lengthy.

  1. As the tests have been conducted in a controlled environment of a laboratory, how it results can be translated to the actual field situation.

  1. Conclusion is poorly drawn – is mainly the repetition of the results which you have already reported in a separate section. Again, I would like to see some practical implications of this research. We do not do the research just of the sake of research or just to publish the paper. You do not let your readers to make their own judgement on its implications as I have some implications coming in my mind. It is important that you report these implications with clarity.

  1. I believe you have not solved the whole problem. There should be something which still need to be investigated further. So, it is important to provide some direction for further research. Also, no research can be 100% perfect. You should have some limitations of your research which you should report here. Do not let your readers to assume the limitations of research. They might devalue your research. So, this should also come from your side.

  1. The paper is under references. Make sure you cite all the relevant research in your paper. I have provided some in my above comments which need to be considered.

  1. The structure could be improved. There are too many figures/tables and I think some of them can be easily removed without any loss. I do not want to be dictating here, so want you to make this decision. Some sentences are also difficult to follow or do not make sense. Please do a through proofread before you submit the revise version.

Author Response

Thank You very much for your favorable review of our manuscript. The responses to the points contained in the review are contained in a separate file.

Round 2

Reviewer 4 Report

Analysis of the aggregate production process with different geometric properties in the light fraction separator

While the authors have made some changes to the paper but there are still few issues that need to be addressed before the paper can be accepted for publication.

1.       Remove any citation from abstract.

2.       The revised paper has become more lengthy which affect the readability. Try to remove unnecessary text/tables/figures to make the paper shorter.

3.       I still do not see strong compelling arguments for this research. apart from the providing the purpose of the research, you need to tell the readers (in a more convincing manner) the reasons/ implications of the proposed research – may be considering practical/industrial and or educational/research implications.

4.       The research approach is still not convincing. It is poorly described with almost no justifications and reference to relevant literature.

5.       There are still some sentences which do not provide clarity. I suggest a thorough proofread.

6.       There should be no citations in the conclusion section. If you have any important citations that should be included in results and analysis section, not in the conclusion section. In the conclusion section you conclude your work not to bring new point, so there is no need for the new citations.

Author Response

Thank You for Your valuable remarks. Please find the answers to your questions in attached file.
